# Plan Your Target and Learn Your Skills: State-Only Imitation Learning via Decoupled Policy Optimization

## Abstract

State-only imitation learning (SOIL) enables agents to learn from massive demonstrations without explicit action or reward information. However, previous methods attempt to learn the implicit state-to-action mapping policy directly from state-only data, which results in ambiguity and inefficiency. In this paper, we overcome this issue by introducing hyper-policy as sets of policies that share the same state transition to characterize the optimality in SOIL. Accordingly, we propose Decoupled Policy Optimization (DPO) via explicitly decoupling the state-to-action mapping policy as a state transition predictor and an inverse dynamics model. Intuitively, we teach the agent to plan the target to go and then learn its own skills to reach. Experiments on standard benchmarks and a real-world driving dataset demonstrate the effectiveness of DPO and its potential of bridging the gap between reality and simulations of reinforcement learning.

## 1 Introduction

Imitation learning offers a way to train an intelligent agent from demonstrations by mimicking the expert's behaviors without constructing hand-crafted reward functions [13, 17]. The corresponding methods normally require the expert demonstrations include information of both states and actions. Unfortunately, the action information is not always accessible from many real-world demonstration resources, e.g., online video recordings of car driving or sports. Thus a natural desire to take advantage of these massive and valuable resources motivates the study of state-only imitation learning (SOIL), also known as learning from observations (LfO) [24]. Analogy to human beings, SOIL is a more intuitive way to approach imitation by only matching the expert's state sequences without having explicit knowledge of the exact actions.

A wide range of algorithms have been proposed to solve SOIL by matching the state sequence of the expert [22, 23, 25]. However, the action agnostic setting in SOIL makes it challenging to determine the optimal action because of the partial observability of the expert demonstrations that multiple policies could be chosen to match the same expert state sequence. Thus learning a state-to-action policy is implicit, leading to a less efficient modeling of the explicit information from demonstrations, and in result could cause suboptimality.

To this end, in this paper, we introduce the concept of *hyper-policy* denoting a *family* of policies that share the same state transition. Based on that, instead of recovering the expert *policy*, we characterize the optimality in SOIL by finding the expert *hyper-policy*. The proposed method is called decoupled policy optimization (DPO), which separates the policy into two modules: an expert state transition predictor that finds the optimal *hyper-policy*, followed by an inverse dynamics model that builds the executable *policy* to deliver actions. Intuitively, the expert state transition predictor predicts the target,

while the inverse dynamics model enables the agent to learn its own skills to reach the target. DPO takes the advantage of such a decoupled structure by separately learning two kinds of data: (1) the expert state transition that is directly accessible in the demonstration; (2) the action to be performed which should be obtained by interacting with the environment.

To ensure the benefit of DPO, these two modules should work coherently to provide accurate foresight for targets and corresponding skills. To achieve this, we regularize the state transition predictor to prevent the model from predicting non-neighboring states via multi-step and cycle training style. Further, to improve the learning efficiency by encouraging the agent to reach the expert states, we augment reward and apply policy gradient to DPO with additional generative adversarial objective.

Experiments on standard benchmarking tasks show the advantage of the decoupled structure and the higher efficiency of DPO. We also evaluate DPO on a real-world driving dataset with state-only demonstrations, and the result shows that DPO can learn driving behaviors closer to human drivers when compared with baseline methods.

## 2 Preliminaries

**Markov Decision Process.** We consider a $\gamma$-discounted infinite horizon Markov decision process (MDP) as a tuple $\mathcal{M} = \langle \mathcal{S}, \mathcal{A}, \mathcal{T}, \rho_0, r, \gamma \rangle$, where $\mathcal{S}$ is the set of states, $\mathcal{A}$ represents the action space, $\mathcal{T} : \mathcal{S} \times \mathcal{A} \times \mathcal{S} \to [0, 1]$ is environment dynamics distribution, $\rho_0 : \mathcal{S} \to [0, 1]$ is the distribution of the initial state $s_0$, and $\gamma \in [0, 1]$ is the discount factor. The agent makes decisions through a policy $\pi(a|s) : \mathcal{S} \times \mathcal{A} \to [0, 1]$ and receives rewards $r : \mathcal{S} \times \mathcal{A} \to \mathbb{R}$.

**Occupancy Measure.** The concept of occupancy measure (OM) [10] is proposed to characterize the statistical properties of a certain policy interacting with an MDP. Specifically, the state OM is defined as the time-discounted cumulative stationary density over the states under a given policy $\pi$: $\rho_\pi(s) = \sum_{t=0}^{\infty} \gamma^t P(s_t = s|\pi)$. Following such a definition we can define different OM:

a) State-action OM: $\rho_\pi(s, a) = \pi(a|s)\rho_\pi(s)$

b) State transition OM: $\rho_\pi(s, s') = \int_{\mathcal{A}} \rho_\pi(s, a)\mathcal{T}(s'|s, a) \, \mathrm{d}a$

c) Joint OM: $\rho_\pi(s, a, s') = \rho_\pi(s, a)\mathcal{T}(s'|s, a)$

**Imitation Learning from State-Only Demonstrations.** Imitation learning (IL) [13] studies the task of learning from demonstrations (LfD), which aims to learn a policy from expert demonstrations without getting access to the reward signals. The expert demonstrations typically consist of expert state-action pairs. General IL objective minimizes the state-action OM discrepancy:

$$\pi^* = \arg\min_\pi \mathbb{E}_{s \sim \rho_\pi^s} \left[ \ell\left(\pi_E(\cdot|s), \pi(\cdot|s)\right) \right] \Rightarrow \arg\min_\pi \ell\left(\rho_{\pi_E}(s, a), \rho_\pi(s, a)\right) , \tag{1}$$

where $\ell$ denotes some distance metric. For example, GAIL [10] chooses to minimize the JS divergence $\mathrm{D}_{\mathrm{JS}}(\rho_{\pi_E}(s, a) \| \rho_\pi(s, a))$, and AIRL [5] utilizes the KL divergence $\mathrm{D}_{\mathrm{KL}}(\rho_{\pi_E}(s, a) \| \rho_\pi(s, a))$ instead, which corresponds to a maximum entropy solution with the recovered reward [17]. However, for the scenario studied in this paper, the action information is absent in state-only demonstrations, known as state-only imitation learning (SOIL) or learning from observations (LfO) problems, where the action spaces between the expert and the agent can even be different. Such challenges prevent applying typical IL solutions. An existing method for this problem is to instead optimize the discrepancy of the state transition OM with the state-to-action policy $\pi(a|s)$ [23]:

$$\pi^* = \arg\min_\pi [\ell\left(\rho_{\pi_E}(s, s'), \rho_\pi(s, s')\right)] . \tag{2}$$

However, the solution to this problem is ambiguous since there is no one-to-one correspondence between $\rho(s, s')$ and $\pi$ as we will show in the following section. As such, the optimality of SOIL should be reconsidered.

## 3 Methodology

### 3.1 Characterizing the Optimality in SOIL

In standard IL tasks, when the expert actions are accessible in demonstrations, perfectly imitating the expert policy corresponds to matching the state-action OM due to the one-to-one correspondence

between $\pi$ and $\rho_\pi(s,a)$ [10, 21]. However, such correspondence is not applicable for the state transition OM matching in SOIL.

**Proposition 1.** *Suppose $\Pi$ is the policy space and $\mathcal{P} = \{\rho : \rho \geq 0\}$ is a feasible set of OM, then a policy $\pi \in \Pi$ corresponds to one state transition OM $\rho_\pi \in \mathcal{P}$. However, a state transition OM $\rho \in \mathcal{P}$ can correspond to more than one policy in $\Pi$.*

The proof can be found in Appendix B. As a result, if we choose to optimize a state-to-action mapping policy, then the optimal solution to Eq. (2) is ambiguous. The ambiguity also comes from the fact that Eq. (2) does not correspond to a maximum policy entropy solution as in normal IL tasks (see Appendix C for details). Therefore, a state-to-action mapping function may be too implicit for matching the state sequence, which could cause training instability and lead to sub-optimal policies. In that case, we must find a one-to-one corresponding solution to solve SOIL explicitly and efficiently. Before continuing, we introduce the definition of *hyper-policy*.

**Definition 1.** *A hyper-policy $\Omega$ is a set of policies such that for any $\pi_1, \pi_2 \in \Omega$, we have $\rho_{\pi_1}(s,s') = \rho_{\pi_2}(s,s')$.*

Then by definition, there is a one-to-one correspondence between the hyper-policy $\Omega$ and the state transition OM $\rho_\Omega(s,s')$. Similar to the normal state-to-action mapping policy, a hyper-policy $\Omega$ can be regarded as a state-to-state mapping function $h_\Omega(s'|s)$ which predicts the state transition such that for any $\pi \in \Omega$:

$$h_\Omega(s'|s) = \frac{\rho_\Omega(s,s')}{\int_{\tilde{s}} \rho_\Omega(s,\tilde{s})\,\mathrm{d}\tilde{s}} = \int_a \pi(a|s)\mathcal{T}(s'|s,a)\,\mathrm{d}a \ . \tag{3}$$

**Proposition 2.** *Suppose the state transition predictor is defined as in Eq. (3) and $\Gamma$ is its space, $\mathcal{P} = \{\rho : \rho \geq 0\}$, then a hyper-policy state transition predictor $h_\Omega \in \Gamma$ corresponds to one state transition OM $\rho_\Omega \in \mathcal{P}$; and a state transition OM $\rho \in \mathcal{P}$ only corresponds to one hyper-policy state transition predictor such that $h_\rho = \rho(s,s')/\int_{\tilde{s}} \rho(s,\tilde{s})\,\mathrm{d}\tilde{s}$.*

Therefore, we find a one-to-one correspondence between the optimization term $\rho(s,s')$ and a practical target $h_\Omega(s'|s)$, which indicates that we do not have to infer the expert actions under state-only demonstrations but only need to recover the state transition predictor of the hyper-policy $\Omega_E$:

$$\arg\min_\Omega[\ell\,(\rho_{\Omega_E}(s,s'), \rho_\Omega(s,s'))] \Rightarrow \arg\min_{h_\Omega} \mathbb{E}_{s\sim\Omega}[\ell\,(h_{\Omega_E}(s'|s), h_\Omega(s'|s))] \ . \tag{4}$$

However, SOIL still requires to learn a policy to interact with the MDP environment to match the state transition OM of the expert. This is achievable since we do not have to recover the expert policy $\pi_E$ exactly but can learn any policy $\pi \in \Omega_E$ according to Eq. (4).

## 3.2 Policy Decoupling

To construct an unambiguous objective for SOIL, we define hyper-policy and solve the problem by finding the state transition predictor of the expert hyper-policy. Intuitively, this tells the agent the *target* that the expert will reach without informing any feasible *skill* that require the agent to learn itself. Therefore, to recover a $\pi \in \Omega_E$, we can construct an inverse dynamics such that

$$\pi = \underbrace{\mathcal{T}_\pi^{-1}}_{\text{Inverse dynamics}} (\ \underbrace{\mathcal{T}(\pi_E)}_{\text{Expert state transition predictor}}\ ) \ . \tag{5}$$

Formally, the expert policy can be decoupled as

$$\pi_E(a|s) = \int_{s'} \mathcal{T}(s'|s,a)\pi_E(a|s)\,\mathrm{d}s' = \int_{s'} \frac{\rho_{\pi_E}(s,a,s')}{\rho_{\pi_E}(s)}\,\mathrm{d}s' = \int_{s'} \frac{\rho_{\pi_E}(s,s')I_{\pi_E}(a|s,s')}{\rho_{\pi_E}(s)}\,\mathrm{d}s'$$
$$= \int_{s'} h_{\pi_E}(s'|s)I_{\pi_E}(a|s,s')\,\mathrm{d}s' \ . \tag{6}$$

Notice that both the state transition predictor $h$ and the inverse dynamics model $I$ is policy dependent. Nevertheless, recall that the optimality in SOIL only requires us to recover $\pi \in \Omega_E$, we do not have to learn about $I_{\pi_E}$ but just one feasible skill $I(a|s,s')$. Then a policy can be recovered by

$$\pi = \mathbb{E}_{\underbrace{s' \sim h_{\Omega_E}(s'|s)}_{\text{target}}}\Big[\underbrace{I(a|s,s')}_{\text{skill}}\Big] \ . \tag{7}$$

Here the inverse dynamics model $I$ offers an arbitrary *skill* to reach the expected *target* state provided by the state transition predictor $h$. In fact, it does not depend on the hyper-policy $\Omega_E$ but a sampling policy $\pi_\mathcal{B}$ to construct $I = I_{\pi_\mathcal{B}}$. We only need a mild requirement for $\pi_\mathcal{B}$ that it covers the support of $\rho_{\Omega_E}(s, s')$ so that the learned $I$ can provide a possible action to achieve the target state. In both experiments and theoretical analysis we show that this requirement alleviates the dependence on the inverse dynamics. Furthermore, if the environment and the expert policy are both deterministic (which is usually the case in real-world scenarios such

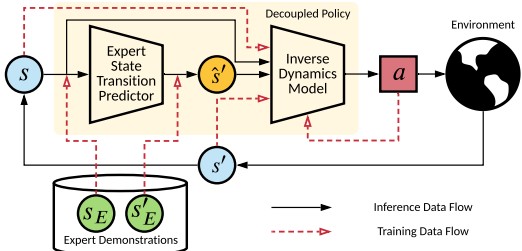

Figure 1: The architecture of Decoupled Policy Optimization (DPO), which consists of an expert state transition predictor (to plan where to go) followed by an inverse dynamics model (to decide how to reach).

as robotics), the state transition is a single-point distribution (or known as the Dirac delta function), and we can simply model $h$ as a deterministic function. By decoupling the policy, which is a state-to-action mapping function, as a state-to-state mapping function (the transition predictor) and a state-pair-to-action mapping function (the inverse dynamics model), we can mimic the expert policy from state-only demonstrations by optimizing these two modules. The whole architecture is illustrated in Fig. 1.

**State Transition Predictor.** In practice, we construct a parameterized expert state transition predictor $h_\psi$ which predicts the subsequent state of the expert taking the input as a current state $\hat{s}' = h_\psi(s)$.

The state transition predictor models the explicit information of the expert, and it can be learned from the demonstration data only. Thence, we implement Eq. (4) as a KL divergence minimization:

$$\min_\psi \mathbb{E}_{(s,s')\sim\Omega_E}[D_{\text{KL}}(h_{\Omega_E}(s'|s)\|h_\psi(s'|s))] \,, \tag{8}$$

which can be optimized in a supervised manner. Specifically, we sample state transitions $(s, s')$ from the expert demonstrations $\mathcal{D}$ and optimize the L2 loss:

$$\mathcal{L}_\psi^h = \mathbb{E}_{(s,s')\sim\mathcal{D}}\left[\|s' - h_\psi(s)\|^2\right] \,. \tag{9}$$

**Inverse Dynamics Model.** Knowing where to go is not enough since the agent has to interact with the environment to reach the target. This can be achieved via an inverse dynamics model, which predicts the action given two consecutive states. Formally, let the $\phi$-parameterized inverse dynamics model $I_\phi$ take input the state pair and predict the feasible action to achieve the state transition: $\hat{a} = I_\phi(s, s')$. Intuitively, we want the inverse dynamics to learn from possible transitions sampled by the agent. Recall that we only need the support of learned $I(a|s, s')$ of the sampling policy covers the support of the expert state transition OM, from which we can infer at least one possible action. Hence, we can optimize the KL divergence between the inverse dynamics of a sampling policy $\pi_\mathcal{B}$ and $I_\phi$:

$$\min_\phi \mathbb{E}_{(s,s')\sim\pi_\mathcal{B}}[D_{\text{KL}}(I_{\pi_\mathcal{B}}(a|s, s')\|I_\phi(a|s, s'))] \,, \tag{10}$$

and we can choose to optimize L2 loss in a supervised manner by sampling from the replay buffer $\mathcal{B}$:

$$\mathcal{L}_\phi^I = \mathbb{E}_{(s,a,s')\sim\mathcal{B}}\left[\|a - I_\phi(s, s')\|^2\right] \,. \tag{11}$$

In our implementation, both the state predictor and the inverse dynamics can be constructed as Gaussian distributions similar to a normal stochastic policy, thus encouraging exploration.

### 3.3 Tackling Compounding Error Challenges

In our formulation, we have decoupled the state-to-action mapping policy as a state-to-state mapping function and a state-pair-to-action mapping function. Unfortunately, the compounding error problem exists such that the agent cannot reach where it plans due to the fitting errors of these two parts.

**Theorem 1** (Error Bound of DPO). *Consider a deterministic environment whose dynamics transition function $\mathcal{T}(s, a)$ is deterministic and L-Lipschitz. Assume the ground-truth state transition $h_{\Omega_E}(s)$ is deterministic, and for each policy $\pi \in \Pi$, its inverse dynamics $I_\pi$ is also deterministic and*

164 *C-Lipschitz. Then for any state s, the distance between the desired state $s'_E$ and reaching state $s'$*
165 *sampled by the decoupled policy is bounded by*

$$\|s' - s'_E\| \le LC\|h_{\Omega_E}(s) - h_\psi(s)\| + L\|I_{\pi_B}(s, \hat{s}') - I_\phi(s, \hat{s}')\| , \tag{12}$$

166 *where $\pi_B$ is a sampling policy that covers the state transition support of the expert hyper-policy and*
167 *$\hat{s}' = h_\psi(s)$ is the predicted next state.*

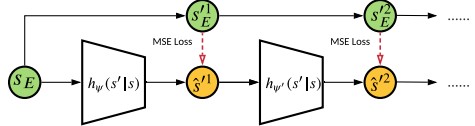

168 The proof can be found in Appendix B, where we also
169 induce a similar error bound for rollout with a state-
170 to-action policy as BCO [22] to show the advantage
171 of the decoupled structure. From Theorem 1 we
172 know that the compounding error can be enlarged
173 due to each part's fitting error, where the first term
174 corresponds to the error of predicted states and the
175 second term indicates whether the agent can reach
176 where it plans to. To alleviate the error, we further propose regularization on these two modules.

Figure 2: Multi-step optimization. Given an expert state $s_E$, $h_\psi$ predicts the next possible state $\hat{s}'^1$, which is further fed to a target network $h_{\psi'}$ to predict the following sequence. The total loss computes the MSE loss along the state sequence.

### 3.3.1 Regularization on Target Planning

178 One major problem is that the state transition predictor may suggest non-neighboring states instead
179 of predicting one-step reachable states. To overcome this, we draw inspiration from Asadi et al. [2]
180 and Edwards et al. [3], and regularize state transition predictor to prevent the model from predicting
181 non-neighboring states via multi-step and cycle training style.

182 **Multi-Step Optimization.** We first explain the details of the multi-step optimization objective.
183 This idea is motivated by Asadi et al. [2], which optimizes a multi-step outcome by executing a
184 sequence of actions in the dynamics model. Here we optimize the state sequence instead. As shown
185 in Fig. 2, given an expert state $s_E$, $h_\psi$ predicts the next possible state $\hat{s}'$ that the expert will reach; the
186 predicted state is then fed into the predictor to output the predicted two-step state $\hat{s}''$. As such, the
187 multi-step training loss is the L2 loss computed along the $k$-step outcome sequence:

$$\mathcal{L}_\psi^{h,\text{ms}} = \mathbb{E}_{(s, \{s'^i_E\}_{i=1}^k) \sim \mathcal{D}} \left[ \|s'^1_E - h_\psi(s)\|^2 + \sum_{i=2}^k \|s'^i_E - h_{\psi'}(s'^{i-1})\|^2 \right] . \tag{13}$$

188 Intuitively, such a regularization makes the state prediction
189 $\hat{s}'$ close to the expert state distribution in order to make
190 accurate long step predictions. It is worth noting that the
191 gradient of the cascading state transition predictors should
192 be dropped since we already have an accurate input at each
193 time step, and for each training step, we only update the
194 first one. We use a target network $h_{\psi'}$ in practice.

195 **Cycle Training Style.** Another way to regularize the
196 transition predictor's output to a neighboring state is to
197 keep an additional function to ensure the cycle consistency,
198 which is also an important technique in [3]. In particular,
199 as illustrated in Fig. 3, given an expert state $s_E$, we take
200 the predicted state $\hat{s}'$ and $s_E$ into the inverse dynamics and
201 get the action $a$, then we train an additional forward dynamics model $M_\omega$ to simulate one step rollout
202 that takes the input $(s_E, a)$ and gets a forward next state $\tilde{s}'$:

Figure 3: Cycle training style. Given an expert state $s_E$, $I_\phi(s, s')$ takes input the predicted state $\hat{s}'$ and $s_E$ to get the execution action $a$, then an additional forward dynamics model $M_\omega$ is used to simulated one step rollout using $(s_E, a)$ and get a forward next state $\tilde{s}'$. The total loss computes the MSE loss between the two predicted states.

$$\begin{aligned} \mathcal{L}_\omega^M &= \mathbb{E}_{(s,a,s') \sim \mathcal{B}} \left[ \|s' - M_\omega(s, a)\|^2 \right] \\ \mathcal{L}_\psi^{h,\text{cycle}} &= \mathbb{E}_{(s,s') \sim \mathcal{D}} \left[ \|s' - h_\psi(s)\|^2 + \|h(s) - M_\omega(s, I_\phi(s, s'))\|^2 \right] . \end{aligned} \tag{14}$$

203 In other words, the cycle training scheme provides a regularization on $h_\psi$ to make predictions
204 consistent with the forward dynamics model.

### 3.3.2 Efficient Skills Learning via Decoupled Policy Gradient

In previous sections, we have mentioned that learning to reach a specific place requires the data-collecting policy to cover the support of the expert hyper-policy. This is easy to achieve on simple low-dimensional tasks, but may not be satisfied in high-dimensional continuous environments. To this end, we encourage the agent to approach those state transitions from the expert's hyper-policy $\Omega_E$ by minimizing the JS divergence of the state transition occupancy using a state-to-action mapping policy $D_{JS}\left(\rho_{\pi_E}(s, s'), \rho_{\pi}(s, s')\right)$. This can be done by producing informative rewards via GAN-like methods [10, 23], and updating the decoupled policy with policy gradients (PG).

In detail, we construct a parameterized discriminator $D_{\omega}(s, s')$ to compute the reward $r(s, a) \triangleq r(s, s')$ as $\log D_{\omega}(s, s')$ and the decoupled policy served as the generator. In addition, since we decouple the policy as two parameterized modules, i.e., a state transition predictor and an inverse dynamics model, then by chain rule, the PG for the decoupled policy can be accomplished by

$$
\begin{aligned}
\nabla \mathcal{L}^{\pi}_{\phi,\psi} &= \mathbb{E}_{\pi}\left[Q(s,a)\nabla_{\phi,\psi}\log \pi_{\phi,\psi}(a|s)\right] \\
&= \mathbb{E}_{\pi}\left[Q(s,a)\int_{s'}\left(\nabla_{\psi}\log h_{\psi}(s'|s) + \nabla_{\phi}\log I_{\phi}(a|s,s')\right)\mathrm{d}s'\right] ,
\end{aligned}
\tag{15}
$$

where $Q$ is the state-action value function; the first term is the gradient for updating the state transition predictor; and the second term is for the inverse dynamics model. Thus, the optimization for both the state transition predictor and the inverse dynamics model can augment the supervised learning objectives with any PG-based learning algorithms (e.g., TRPO, PPO, SAC). As the training proceeds, the agent will sample more transition data around $\Omega_E$, and thus the support of the sampling policy will progressively cover the support of $\rho_{\Omega_E}(s, s')$.

### 3.4 Overall Algorithm

By combining the idea of generative adversarial training, we obtain our final algorithm, composed with three essential parts: the state transition predictor $h$ used for predicting the possible future states sampled by the expert; the inverse dynamics model $I$ used for inferring the possible actions conditioned on two adjacent states; and the discriminator $D$ used for offering intermediate reward signals for training the decoupled policy $\pi = I(h)$. The overall objective of DPO is

$$
\mathcal{L}^{\pi,h,I}_{\phi,\psi} = \lambda_G \mathcal{L}^{\pi}_{\phi,\psi} + \lambda_h \mathcal{L}^{h}_{\psi} + \lambda_I \mathcal{L}^{I}_{\phi} ,
\tag{16}
$$

where $\lambda_G$, $\lambda_h$ and $\lambda_I$ are hyperparameters for trading off the training among each loss. In practice, we try less than ten combinations for these parameters as shown in Appendix D.3, and we directly optimize $\mathcal{L}^{\pi}_{\phi,\psi}$ instead of iterative training. The detailed algorithm is summarized in Appendix A. Besides, it is worth noting that both the inverse dynamics model and the state transition predictor can be pre-trained, where we optimize $\mathcal{L}^{h}_{\psi}$ using the state-only demonstration and optimize $\mathcal{L}^{I}_{\phi}$ using samples collected by a randomized agent.

## 4 Related Work

State-only imitation learning (SOIL) endows the agent with the ability to learn from expert states. Although lacking the expert decision information, most of the previous works still optimize a state-to-action mapping policy to match the expert state transition distribution. For example, Torabi et al. [22] used a model-based approach to apply behavioral cloning to state-only demonstrations, while Torabi et al. [23] employed a similar structure to GAIL to match the state transition distribution. Yang et al. [25] analyzed the optimization gap between SOIL and naive IL and introduced a mutual information term to narrow it. Huang et al. [11] applied SOIL on autonomous driving tasks by decoupling the policy into a neural decision module and a non-differentiable execution module in a hierarchical way.

Our work decouples the state-to-action policy into two modules. However, both the inverse dynamics model and the state transition predictor have been widely used by many previous works on RL and IL tasks. For instance, Torabi et al. [22] and Guo et al. [6] trained an inverse dynamics model to label the

Table 1: Comparison between different methods.

| Method | Inverse Dynamics | State Predictor | Decoupled Policy | Task |
|---|---|---|---|---|
| BCO [22] | ✓ | ✗ | ✗ | SOIL |
| GAIfO [23] | ✗ | ✗ | ✗ | SOIL |
| IDDM [25] | ✗ | ✗ | ✗ | SOIL |
| OPOLO [27] | ✓ | ✗ | ✗ | SOIL |
| PID-GAIL [11] | ✗ | ✗ | ✓ | IL |
| QSS [3] | ✓ | ✓ | ✓ | RL |
| SAIL [16] | ✓ | ✓ | ✗ | IL |
| DPO (Ours) | ✓ | ✓ | ✓ | SOIL |

Table 2: Eventual performance against different methods on 6 easy-to-hard continuous control benchmarks. The means and the standard deviations are evaluated over more than 5 random seeds.

| | InvertedPendulum | InvertedDoublePendulum | Hopper | Walker2d | HalfCheetah | Ant |
|---|---|---|---|---|---|---|
| Random | $25.28 \pm 5.53$ | $78.28 \pm 10.73$ | $13.09 \pm 0.10$ | $7.07 \pm 0.13$ | $74.48 \pm 12.39$ | $713.59 \pm 203.92$ |
| BCO | $\mathbf{1000.00 \pm 0.00}$ | $415.04 \pm 148.46$ | $1430.16 \pm 398.81$ | $261.36 \pm 25.17$ | $-13.66 \pm 149.94$ | $397.79 \pm 239.16$ |
| GAIfO | $\mathbf{1000.00 \pm 0.00}$ | $7818.07 \pm 1778.67$ | $3068.10 \pm 26.32$ | $3865.20 \pm 341.90$ | $8953.35 \pm 1079.41$ | $5122.29 \pm 807.19$ |
| GAIfO-DP | $\mathbf{1000.00 \pm 0.00}$ | $7305.01 \pm 1591.23$ | $3031.84 \pm 152.13$ | $4003.06 \pm 241.34$ | $8675.42 \pm 807.29$ | $\mathbf{5535.9 \pm 62.74}$ |
| DPO (w/o PG) | $\mathbf{1000.00 \pm 0.00}$ | $3545.70 \pm 738.16$ | $629.84 \pm 344.07$ | $334.23 \pm 85.42$ | $-472.00 \pm 132.81$ | $-196.96 \pm 124.26$ |
| DPO (w PG) | $\mathbf{1000.00 \pm 0.00}$ | $\mathbf{7846.40 \pm 1541.20}$ | $\mathbf{3165.72 \pm 68.44}$ | $\mathbf{4407.53 \pm 266.72}$ | $\mathbf{10501.96 \pm 438.01}$ | $5338.48 \pm 107.2$ |
| Expert (SAC) | $1000.00 \pm 0.00$ | $9358.87 \pm 0.10$ | $3402.94 \pm 446.48$ | $5639.32 \pm 29.97$ | $13711.64 \pm 111.47$ | $5404.55 \pm 1520.49$ |

state-only demonstrations with inferred actions. Nair et al. [19] proposed a method for manipulating ropes to match a single human-specified image sequence, in which an inverse dynamics model is trained in a self-supervised manner and used to generate control signals. Kimura et al. [14] utilized a state transition predictor to fit the state transition probability in the expert data, which is further used to compute a predefined reward function. Liu et al. [16] constructed a policy prior using the inverse dynamics and the state transition predictor, but the policy prior was only used for regularizing the policy network. However, as shown in this paper, the policy can be exactly decoupled as these two parts, which can be uniformly optimized through policy gradient without keeping an extra policy. Edwards et al. [3] estimated $Q(s, s')$ for RL tasks which employs a similar policy form as Eq. (6) and updates the state transition predictor through a deterministic policy gradient similar to DDPG [15]. To sort out the difference between these methods and ours, we summarize the key factors in Tab. 1.

## 5 Experiments

We conduct four sets of experiments to investigate the following research questions:

**RQ1** Is decoupled learning structure superior than state-to-action structure on SOIL tasks?

**RQ2** Does DPO achieve higher efficiency or better performance than baselines on SOIL tasks?

**RQ3** Can agent reach where it plans and does the proposed regularization help mitigate the compounding error?

**RQ4** How can DPO be applied on real-world data?

To answer RQ1, we conduct toy experiments with a simple 2D grid world environment and compare both qualitative and quantitative results of DPO against BCO and GAIfO. Regarding RQ2, we empirically evaluate DPO on easy-to-hard continuous control benchmarking tasks. And for RQ3, we evaluate the difference between the predicted states that the agent plans to reach and the consecutive state that the agent actually reaches in the environment for the proposed regularization. Finally, we try to imitate real-world traffic surveillance recordings in a simulated environment to investigate RQ4, which shows the potential of using real-world data for human behavior simulation. Due to the space limit, we leave experiment details, additional results and ablation study in Appendix.

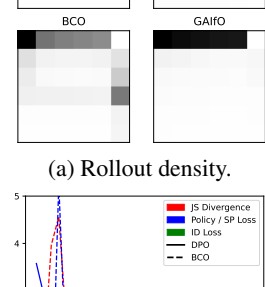

(a) Rollout density.

(b) Loss curves.

Figure 4: Toy example.

### 5.1 Understanding the Decoupled Structure

In this paper we design decoupled policy optimization (DPO) to perform SOIL tasks, and in previous sections we propose that the key technical contribution of DPO is the decoupled structure of policy that models the explicit state transition information and the latent action information from demonstrations, which solves the ambiguity and enhances the learning efficiency. Therefore, in this set of experiments, we aim to demonstrate how DPO is superior than state-to-action policy methods (RQ1). We first generate expert demonstrations in a 2D 6×6 grid world environment, in which the agent starts at the upper left corner and aims to reach the upper right corner. In each grid the agent has $k \times 4$ actions, which means that the agent has $k$ possible actions to reach the neighboring block and in our experiment we choose $k = 5$ to enlarge the action space.

The density of the expert trajectories and the trajectories sampled by different methods are shown in Fig. 4(a). We show that both BCO and GAIfO have troubles in directly learning the implicit action from state-only behaviors. Notably, GAIfO only imitates the major trajectory and omit the other choice and BCO also stucks in the middle right. By contrast, DPO recovers the expert demonstrations

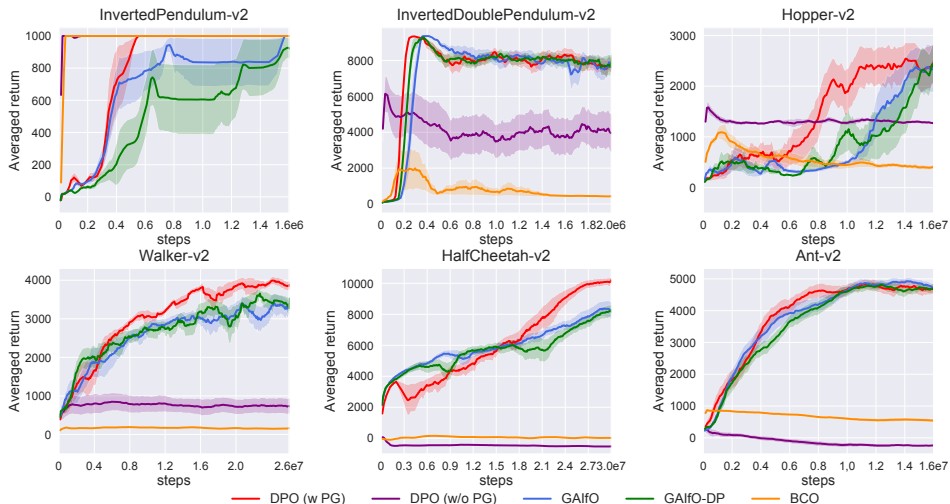

Figure 5: Learning curves on 6 easy-to-hard continuous control benchmarks, where the solid line and the shade represent the mean and the standard deviation of the averaged return over more than 5 random seeds. We pre-train BCO and DPO for 50k steps and show it in figures.

much better, benefiting from the decoupled structure that first determining the target and then taking the action to achieve it. To further illustrate the learning efficiency advantage of DPO, we illustrate the JS divergence curves of DPO and BCO during training in Fig. 4(b). Besides, we show the policy loss for BCO, the state predictor (SP) loss for DPO, and the inverse dynamics (ID) loss for both methods. Except that the JS divergence of DPO decreases more quickly than BCO, it is also observable that DPO relies less on the inverse dynamics than BCO, since the inverse dynamics loss of DPO converges to a higher level. We further provide a theoretic analysis of the dependence on inverse dynamics with BCO and DPO in Appendix B.

## 5.2 Comparative Evaluations

We compare the qualitative results of DPO against other baseline methods on easy-to-hard continuous control benchmarking environments (RQ2), including InvertedPendulum, InvertedDoublePendulum, Hopper, Walker2d, HalfCheetah and Ant. In each environment, besides GAIfO and BCO, we also evaluate GAIfO with decoupled policy (denoted as GAIfO-DP). For DPO we compare the reward augmented version of DPO (denoted as DPO w PG)[1] with the supervised learning version of DPO, i.e., $\lambda_G = 0$ (denoted as DPO w/o PG). For fairness, we re-implement all the algorithms based on a Pytorch code framework[2] and adopt Soft Actor-Critic (SAC) [7] as the RL learning algorithm for GAIfO and DPO. For all environments, we first train an SAC agent to collect 4 state-only expert trajectories and then train agents with such data. All algorithms are evaluated by a deterministic policy. The eventual results are summarized in Tab. 2, and the learning curves are shown in Fig. 5. It is worth noting that for DPO, we choose the best performance among the experiments that use multi-step or cycle regularization, and we put the full experiment results in Appendix D.

One can easily observe that on simple environments, BCO is able to achieve a good performance, and GAIfO also does well on harder tasks. Even so, DPO can still gain the best or comparable performance against its counterparts. Particularly, without augmented reward, DPO is able to reach the optimality with the highest sample efficiency on simple tasks like InvertedPendulum. By contrast, on higher-dimensional tasks such as Hopper, Walker2d, HalfCheetah and Ant, it is difficult to construct accurate inverse dynamics that covers the support of the expert hyper-policy from scratch. However, by combining generative adversarial policy gradients, the agent finally recovers a good policy from the expert hyper-policy. This is particularly evident on HalfCheetah where DPO behaves poorly at the beginning but improves fast as the training proceeds. Besides, as illustrated in Fig. 5, DPO benefits from better sample efficiency in most of the environments, but the improvements are limited on the hardest tasks. We think that this may be due to larger state spaces (111 dimensions for Ant) that makes it difficult to recover a good state predictor or an inverse dynamics model. In all

---

[1]Without ambiguity we simply denote DPO for this version of algorithm in the following sections.
[2]https://github.com/KamyarGh/rl_swiss

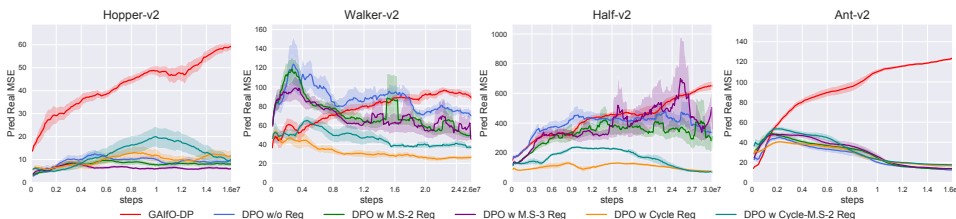

Figure 6: Compounding error of the predicted consecutive states and the real states the agent reaches when rollout in the environments.

experiments, GAIfO-DP achieves similar results as GAIfO, indicating that the network structure does not count much for the performance.

## 5.3 Compounding Error Reduction

In this section, we aim to study whether the agent can effectively reach the target as it plans (RQ3). Therefore, we analyze the distance of the reaching states and the predicted consecutive states, and draw the mean square error (MSE) along the training procedure in Fig. 6. We also compare our regularization including multi-step optimization (denoted as M.S.-$k$, where $k$ is the number of rollout steps) and cycle training style (denoted as Cycle). Note that DPO needs at least 1-step rollout for training the state transition predictor. As shown in Fig. 6, the agent still has gaps to get to where it plans to, and the mismatch always deteriorates on harder tasks. Combining regularization can always achieve lower compounding error, and the cycle training is effective in most of the environments. In Appendix D.6, we further illustrate the correlation between the final performance and the distance.

## 5.4 Learn to Drive from Real-World Traffic Data

The rapid development of autonomous driving has brought a lot of demand for simulating and training an RL agent in the simulator, which requires realistic interactions with various social vehicles [26]. However, driver's detailed actions are not easily to obtain yet we adopt SOIL from a traffic surveillance recording dataset (NGSIM I-80 [8]) that contains kinds of recorded human driving trajectories. We wish to further examine the potential of DPO for decreasing the gap between the real world and simulation (RQ4). We utilize the simulator provided by Henaff et al. [9] as our simulation platform and learn to imitate real-world driving behaviors. We compare DPO against GAIfO and BCO, and choose *Success Rate*, *Mean Distance* and *KL Divergence* as evaluation metrics. Specifically, *Success Rate* is the percentage of driving across the entire area without crashing into other vehicles or driving off the road, *Mean Distance* is the distance traveled before the episode ends, and *KL Divergence* measures the position distribution distance between the expert and the agent.

As shown in Tab. 3, DPO outperforms baseline methods in all three metrics while possessing higher stability. The decoupled policy allows the state predictor to focus on matching the distribution of expert trajectories, thus achieving smaller deviations from the expert position distribution. Furthermore, since the policy gradient can be computed with non-differentiable inverse dynamics, we can generate stable action sequences [12, 11] by replacing the inverse dynamics model with classical controllers, which can be generalized to realistic applications.

Table 3: Performance on NGSIM I-80 driving task over 5 random seeds.

| Method | Success Rate (%) | Mean Distance (m) | KL Divergence |
|---|---|---|---|
| BCO | $27.4 \pm 1.1$ | $129.8 \pm 2.0$ | $24.4 \pm 2.2$ |
| GAIfO | $77.5 \pm 0.8$ | $188.3 \pm 1.1$ | $11.5 \pm 3.9$ |
| DPO | $\mathbf{80.3 \pm 0.5}$ | $\mathbf{192.7 \pm 0.6}$ | $\mathbf{9.5 \pm 1.8}$ |
| Expert | 100 | 210.0 | 0 |

## 6 Conclusion

In this paper, we characterize the optimality and investigate the ambiguity problem in state-only imitation learning, and accordingly propose Decoupled Policy Optimization (DPO), which splits the state-to-action mapping policy into a state-to-state mapping state transition predictor and a state-pair-to-action mapping inverse dynamics model. Furthermore, we employ regularization and generative adversarial methods to mitigate the compounding error caused by the decoupled modules. The flexibility of the decoupled architecture allows a wide range of interesting future works, such as replacing the inverse dynamics with a classic control module to produce stable control signals, learning specific skills with shared state transition and multi-task target learning with shared pre-trained skills.

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
