# OpenReview forum: "Plan Your Target and Learn Your Skills: State-Only Imitation Learning via Decoupled Policy Optimization"
_NeurIPS.cc/2021/Conference — NeurIPS 2021 Submitted_

### Official Review · Reviewer_M4Kk · 2021-07-11

**Rating:** 5
**Confidence:** 4

**Summary:**

The paper proposes a new method for state-only imitation learning, based around explicitly representing the transition from one state to the next (based on state-only expert demonstrations) and an inverse dynamics model providing feasible actions to produce that transition (based on rollouts from a sampling policy). Theoretical analysis attempts to show that this reduces policy ambiguity. Empirical results show improved performance relative to baselines, with the most striking results in a gridworld and on real-world driving data: on standard MuJoCo continuous control tasks, the mean of the method tends to be higher, but the confidence intervals overlap with baselines.

**Limitations And Societal Impact:**

There is no discussion of social impact, and little discussion of limitations -- there is a brief discussion of future work in the conclusion.

**Main Review:**

There are many data sources that only provide observations. To exploit such data sources, we need to be able to learn from data without actions (the focus of this paper) and from partially observable data (not the focus). The basic idea of exploiting the structure of the problem by separately learning to predict the expert state transitions and to predict actions that lead to them is plausible. Unfortunately, the paper is confusingly written, key theoretical details are either incorrect or stated too informally to verify, and the empirical results while promising are not fully convincing. Given these deficiencies, I do not think this manuscript is suitable for presentation at NeurIPS.

# Issues with derivation

The proof for Proposition 1 (appendix B) is very informal. While the overall claim seems approximately correct, it is missing some key assumptions (which you would likely have noticed had the proof been written rigorously). In general one may not be able to recover the policy from the state-action OM, since the OM $\rho(s,a)$ may be zero for all actions $a$ transitioning out of a particular state $s$. For example, this can occur if $s$ is not reachable from the starting state of the MDP. Granted you may not care about this unindentifiability (although it does matter in the transfer case), but it does mean there is policy ambiguity even in the state-action case, and there is no one-to-one correspondence. Additionally, the “proof” that there is no one-to-one correspondence between $\rho(s,s')$ and $\pi(a\mid s)$ seems to be missing a step – simply writing down the marginal formula does not establish this. Indeed, there exist MDPs where the transition dynamics are invertible, so that we can establish $(s,a)$ from $(s,s')$.

Definition 1 is also rather imprecise. A literal reading implies that a hyper-policy is any set of policies where all policies have the same state transition OM. This would include, of course, the empty set and all singleton sets of policies. In this case the following claim that “there is a one-to-one correspondence between the hyper-policy $\Omega$ and the state transition OM $\rho_{\Omega}(s,s')$ is clearly false. For any $\Omega$ with size $|\Omega| > 2$, any subset of it is still a hyper-policy and has the same OM. I think you need the hyper-policy to be the set of *all* policies with this OM for this claim to go through.

Proposition 2 is stated without proof, and I do not believe it to be true due to the issue discussed with the Definition, although the intuition behind it seems plausible so I imagine this can be fixed.

# Experimental results

I appreciated the gridworld experiment -- although simple, this does provide a striking qualitative illustration of the difference between the methods. However, it would help to have some discussion as to why your method performs better in this case. In particular, I suspect the GAIfO failure mode was due to mode collapse, which often happens with GAIL (and GANs more generally). Should we expect your algorithm to systematically avoid these problems, or will similar problems occur given enough task complexity?

Table 2: It appears the confidence interval for DPO overlaps with the confidence interval for at least one other method in every column. Although the means being >= other methods in 5/6 cases is certainly promising, it would be more convincing if the difference was statistically significant. Can you run more seeds to reduce variance?

# Minor points

Correctness:

- “Furthermore, if the environment and the expert policy are both deterministic (which is usually the case in real-world scenarios such as robotics)” – this seems to be claiming that human demonstrations are deterministic. While humans may be deterministic in a philosophical sense, unless you are observing all relevant state (how much sleep the human had, how caffeinated they are, etc) I very much doubt this will be true in practice.

- “we also induce a similar error bound for rollout with a state-to-action policy as BCO [22] to show the advantage of the decoupled structure”. How is the decoupled structure an advantage when the error bound is similar (not stricly less)?

Clarity:

- Line 26: “state-to-action policy” – don't all policies mapping states to (distributions) over actions?

- Line 30: “state transition” is unclear. Do you mean visitation distribution? State sequence?

- Line 89-90: I did not understand any of this: “Therefore, a state-to-action mapping function may be too implicit for matching the state sequence, which could cause training instability and lead to sub-optimal policies. In that case, we must find a one-to-one corresponding solution to solve SOIL explicitly and efficiently.”

- Line 99: “and $\Gamma$ is its space” – what does this mean? Clarity is always important, but is absolutely essential when stating definitions and theorems.

- Would be helpful to introduce BCO somewhere as it's referred to repeatedly in the experiments. The only place it is defined seems to be in a table in related work.

# Update after rebuttal

I would like to thank the authors for a detailed response to my and others reviews. The propositions are now much more clearly stated and although I did not have time to check every detail of the derivation for Proposition 1 it appears to now be sound. In light of this, I am increasing my score.

Thank for you running the additional experiments. In particular it is helpful to see from Figure 3 that BCO learns the correct behavior (albeit slowly) when there is an invertible transition dynamics. I may be willing to increase my score further depending on the confidence intervals (see question below).

**Time Spent Reviewing:**

1.5

---

> ### Author Response · Authors · 2021-08-10
> **Response to Reviewer M4Kk**
>
> Thanks for your detailed comments. We notice that your main criticism focus on the theoretical part, especially details in the proposition, the theorem, or the derivation, instead of the motivation and the novelty of the method itself.  Thus, we will first take our effort to resolve your concerns about the mentioned non-rigorous theory part.
>
> ----------------
> **About the derivation and theoretical concerns.**
> ----------------
>
> We agree that most of the issues you pointed out are right and worthy, however, none of them is fatal incorrect and could be fixed in this rebuttal session. Below we list how we refine these parts as you suggest.
>
> **Q1: "The proof for Proposition 1 (Appendix B) is very informal...it is missing some key assumptions" "In general one may not be able to recover the policy from the state-action OM, since the OM..." "...correspondence between $\rho(s,s')$ and $\pi(a|s)$ seems to be missing a step...where the transition dynamics are invertible."**
>
> **A1:** We agree and value your advice. As our response A4 to reviewer GQAi, we have made a complete proof following the theorem in [2] for this proposition (can be seen in the external link (https://anonymous.4open.science/r/dpo_rebuttal-2AF4/dpo_rebuttal_extra_fig.pdf)).
>
> In addition, for your key concerns about the unrecoverablility of the state-to-action occupancy, we apologize for missing an important condition to explain the "feasible set". In fact, this can be referred to the Lemma 3.1 of [1]. The revised proposition is pasted below.
>
> - Suppose $\Pi$ is the policy space and $\mathcal{P}$ is a valid set of state transition OMs such that $\mathcal{P}= \lbrace\rho:\rho\geq 0\text{ and }\exists\pi\in\Pi, \text{s.t.} \rho(s, s')=\rho_0(s)\int_{a} \pi(a|s)\mathcal{T}(s'|s, a){\mathop{}\mathrm{d}} a+\int_{s'', a} \pi(a|s)\mathcal{T}(s'|s, a)\rho(s'', s){\mathop{}\mathrm{d}} s''{\mathop{}\mathrm{d}}a \rbrace$, then a policy $\pi\in\Pi$ corresponds to one state transition OM $\rho_{\pi}\in\mathcal{P}$. However, under the action-redundant assumption about the dynamics $\mathcal{T}$, a state transition OM $\rho\in\mathcal{P}$ can correspond to more than one policy in $\Pi$.
>
> Furthermore, we now supplement an assumption in Sec 2 about the environmental dynamics for consistency as below.
>
> - For analyzing the effect of action ambiguity, we assume the environment dynamics $\mathcal{T}(s'|s,a)$ has redundant actions whose transition probability can be written as linear combination of other actions. Formally, this refers to the existence of a state $s_m\in\mathcal{S}$, an action $a_n\in\mathcal{A}$ and a distribution $p$ defined on $\mathcal{A}\setminus\lbrace a_n\rbrace$ such that $\int_{\mathcal{A}\setminus \{a_n\}}p(a)\mathcal{T}(s'|s_m, a){\mathop{}\mathrm{d}} a=\mathcal{T}(s'|s_m, a_n)$.
>
> **Q2: "Definition 1 is also rather imprecise."**
>
> **A2:** We appreciate your detailed advice. We follow your suggestion and revise the definition as: "A hyper-policy $\Omega\in\Lambda $ is a **maximal** set of policies sharing the same state transition occupancy such that ..."
>
> **Q3: "Line 99: 'and $\Gamma$ is its space'- what does this mean?" "Proposition 2 is stated without proof, and I do not believe it to be true due to the issue discussed with the Definition"**
>
> **A3:** Thanks for the comment.
> - We have revised the whole proposition 2 to make it clearer as:
>     - Suppose the state transition predictor $h_{\Omega}$ is defined as in Eq. 3 of the main paper and $\Gamma=\lbrace h_{\Omega}: \Omega \in\Lambda\rbrace$ is a valid set of the state transition predictors, $\mathcal{P}$ is a valid set of the state-transition OMs defined as in Proposition 1, then a state transition predictor $h_{\Omega}\in\Gamma$ corresponds to one state transition OM $\rho_{\Omega}\in\mathcal{P}$; and a state transition OM $\rho\in\mathcal{P}$ only corresponds to one hyper-policy state transition predictor such that $h_{\rho} = \rho(s,s')/\int_{s'}\rho(s,s'){\mathop{}\mathrm{d}} s'$.
> - Besides, for completeness, we add a direct proof for proposition 2 in the above link.
>
> We hope our responses and revisions can ease your concern about our derivations.
>
> ----------------
> **About the experiment.**
> ----------------
>
> **Q4: "I suspect the GAIfO failure mode was due to mode collapse, which often happens with GAIL (and GANs more generally). Should we expect your algorithm to systematically avoid these problems, or will similar problems occur given enough task complexity?"**
>
> **A4:** We have supplemented additional experimental results for the grid world in order to clarify this problem which will be put in the revision. The figures of the additional results are also shown in the above link.
> - We first let $k=1$. From the results, we can see that BCO and DPO share similar asymptotical performance (KLD), but DPO achieves a significantly faster convergence rate. On the contrary, GAIfO still fails to find the second path, indicating the mode collapse problem.
> - We complement additional figures (figure 1 & 2 in the external link)  to demonstrate the properties of both decoupled policy modules. We can see that the state transition predictor exactly aligns with the expert state transition (figure 2). More importantly, the action distribution plot indicates that the learned inverse dynamics is different from the expert's, and (almost) equally distributed on ambiguous actions (figure 1). This supports our claim that any inverse dynamics valid on the expert transition support can be used to construct the expert hyper policy, and our algorithm does not exhibit any preference on a particular inverse dynamics.
> - For complex tasks, it is hard to analyze whether DPO also suffers from the mode collapse, but in toy experiments, we can see that additional supervision is possible to ease such problems, and both the final performance and the training efficiency benefit from it.
>
> **Q5: "DPO overlaps with the confidence interval for at least one other method in every column...it would be more convincing if the difference was statistically significant. Can you run more seeds to reduce variance?"**
>
> **A5:** Thanks for your suggestion. Follow your suggestion we conduct evaluation experiments with more seeds (the overall results are updated as shown in the above link that is almost the same). Particularly, for InverseDoublePendulum we bold all overlapped methods. For hopper, walker, and halfcheetah, we conduct a t-test to evaluate whether our method is statistically the best. The p-values against various baselines are shown below.
>
> p-value table of DPO against baselines
>
> | Env | Baseline | p-value |
> |:----------:|:----------:|:----------:|
> | Hopper | BCO  | 7.2e-09 |
> | Hopper | GAIfO  | 6.93e-10 |
> | Hopper | GAIfO-DP  | 2.29e-05 |
> | Walker | BCO  | 2.48e-24 |
> | Walker | GAIfO  | 4.01e-06 |
> | Walker | GAIfO-DP  | 0.000146 |
> | HalfCheetah | BCO  | 4.22e-30 |
> | HalfCheetah | GAIfO  | 0.000206 |
> | HalfCheetah | GAIfO-DP  | 5.35e-06 |
>
> Since all p-values are less than 0.05 and even are less than 0.01, we can say that our method is statistically significantly the best.
>
> ----------------
> **About minor clarification.**
> ----------------
>
> **Q6: "this seems to be claiming that human demonstrations are deterministic. While humans may be deterministic in a philosophical sense, unless you are observing all relevant states... I very much doubt this will be true in practice."**
>
> **A6:**  We appreciate your detailed comments, and we revise the sentence in a more proper way: "Furthermore, if the environment and the expert policy are both deterministic (which is usually the case in **lots of** real-world scenarios such as robotics)".
>
> **Q7: "How is the decoupled structure an advantage when the error bound is similar (not strictly less)?"**
>
> **A7:** In the toy experiment, we observe less dependence on the inverse dynamics of DPO. Although the bound of DPO can not be theoretically showed strictly less than BCO, we analyze it from the factors of those bounds, as in both Sec 5.1 and Appendix B.
>
> Besides, such a decoupled structure shows promising future works to do, such as pretraining the inverse dynamics to tackle multi-task problems; or reusing the state-predictor for transfer learning for agents with different action spaces in the same state space.
>
> **Q8: "Line 26: 'state-to-action policy' – don't all policies mapping states to (distributions) over actions?"**
>
> **A8:** This refers to a state-to-action mapping policy in a non-decoupled structure that maps state to action directly. Our decoupled policy can be regarded as a "state-to-state-to-action" policy.
>
> **Q9: "Line 30: 'state transition' is unclear."**
>
> **A9:** Thanks for pointing this out. We were referring to the state transition occupancy without bringing the notation into the introduction. And we have changed the statement to "state-state visitation distribution" in our revision.
>
> **Q10: "Line 89-90: I did not understand any of this..."**
>
> **A10:** Sorry for the confusion. This explains why the non-decoupled policy is not a good inductive bias in SOIL tasks since we do not have any action information of the expert. And the decoupled structure resolves it by introducing the state transition predictor, which is consistent with the expert data.
>
> **Reference:**
>
> [1] Ho J, Ermon S. Generative adversarial imitation learning[J]. Advances in neural information processing systems, 2016, 29: 4565-4573.
>
> [2] Syed U, Bowling M, Schapire R E. Apprenticeship learning using linear programming[C]//Proceedings of the 25th international conference on Machine learning. 2008: 1032-1039.

---

> > ### Author Response · Authors · 2021-08-10
> > **Response to Reviewer M4Kk (Cont.)**
> >
> > **Other comments:**
> >
> > We would like to thank you again for your comments on the mathematical derivation. We think your suggestions are very useful and constructive for us to polish and improve our paper. While your comments have been focused on theoretical proof details, we believe we have addressed the problems you have raised. We agree that these derivation issues are important yet addressable. We would expect that these concerns have been largely fixed in this rebuttal session and we hope the fixed version won't make the paper "confusingly written" any longer.
> >
> > Thus, we think these concerns may not outweigh the technical contributions to the SOIL problem, and we have clarified most of them to make the paper clearer and rigorous. We sincerely hope you re-evaluate our work. We would be happy to make further corrections if necessary and look forward to your feedback. We commit to further improve the clarity of our paper in the revision.
> >
> > Finally, we will continue to polish our language and the clarity in our revision. Thanks again for your valuable comments.

---

> > ### Comment · Reviewer_M4Kk · 2021-08-13
> > **Updated review & query about p-values**
> >
> > I have updated my original review and increased my score to 4.
> >
> > Thanks for increasing the number of seeds used for evaluation, this will certainly help understand the strengths of this method. However, I am puzzled by the p-values you report. The updated Table 1 has, for example, Hopper with $3163.74 \pm 64.46$ for DPO and $3030.7 \pm 139.85$ for GAIfO-DP which still overlap (and I'll note the GAIfO-DP is not bolded). Yet you list a $p$-value of 2.29e-5. I assume the statistical test and confidence intervals are being computed in a different way. Can you explain the methodology used and why this leads to such a seemingly large discrepancy?

---

> > > ### Author Response · Authors · 2021-08-14
> > > **Additional response for confidence interval**
> > >
> > > Thanks for your feedback!
> > >
> > > Before answering your question, we guess the "confidence interval" you mentioned is the term behind the $\pm$ symbol. In fact, this is the std of the results across different seeds rather than the confidence interval of means, as said in the paper and do in many of the previous imitation learning works [1,2].
> > >
> > > **Short response for you**: (i) The confidence interval is determined by confidence level, the number of samples, and standard deviation; (ii) the 95%-level confidence intervals of these two methods are not overlapped with a sufficient number of samples; (iii) the p-values are getting smaller as the number of samples increases.
> > >
> > > Intuitively, this fact makes more sense after observing that DPO gets a higher mean and a lower std than the other baselines.
> > >
> > > To get an accurate conclusion, we first try to clarify the relationship between std and confidence interval.
> > >
> > > Importantly, to compute the confidence interval, we have to specify an associated probability (1-$\alpha$) as the **confidence level** (e.g., 95%.), and the number of samples $n$. Let's assume our data as Gaussian distribution ($\mu$, $\sigma$), then the $(1-\alpha)$ confidence interval has a form as:
> > > $\mu-c\frac{\sigma}{\sqrt{n}}, \mu+c\frac{\sigma}{\sqrt{n}}$ where $c$ is a constant related to $\alpha$. For example, when $\alpha=0.05$, $c=1.96$. Intuitively, more samples will lead to a smaller CI.
> > >
> > > In our test case, the sample number $n$ is 35 (the number of different seeds in our experiment), then we have $c\frac{1}{\sqrt{n}}\approx 0.33$. From our results, the 95%-level CI for different methods is then $\mu-0.33 \sigma, \mu+0.33 \sigma$. As a result, the 95%-level CI of GAIfO-DP on Hopper is then $[2984.55, 3076.85]$ and that of DPO is then $[3142.47, 3185.01]$, and **there is no overlap between these two CIs** from 95% confidence level.
> > >
> > > Therefore, we can notice that $mean\pm std$ results actually do not reflect whether the confidence intervals are overlapped and the number of samples counts a lot. In our evaluation, with less data, the p-value just shows slight significance (in that case the confidence intervals also overlap). However, **with more evaluated data, the difference between the two data sets is more significant, which results in a lower p-value**, this is consistent with the CI.
> > >
> > > The discussion with you has been enlightening to us, which led us to think deeper about these statistics. The confidence interval can better distinguish between different methods in some cases and takes into account the uncertainty associated with the amount of data sampled. Therefore, we consider adding a table of mean values and their confidence intervals of different methods in the Appendix.
> > >
> > >
> > > References:
> > >
> > > [1] Ho J, Ermon S. Generative adversarial imitation learning[J]. Advances in neural information processing systems, 2016, 29: 4565-4573.
> > >
> > > [2] Ghasemipour S K S, Zemel R, Gu S. A divergence minimization perspective on imitation learning methods[C]//Conference on Robot Learning. PMLR, 2020: 1259-1277.
> > >
> > > For your reference to test on your own, we provide you the re-run evaluation results. You can pick the top 5 results of each method and the whole results to compute the mean/std and p-value for comparison.
> > > Reference code and values for conducting t-test:
> > >
> > > ```python
> > > from scipy import stats
> > >
> > > var_homo = stats.levene(baseline_array, your_method_array)
> > > flag = var_homo.pvalue > 0.05 # same variance flag
> > > res = stats.ttest_ind(baseline, your_method, equal_var=flag)
> > > print(res.pvalue)
> > > ```
> > >
> > > ```python
> > > GAIfO-DP=[
> > > 3054.016211,
> > > 3095.186422,
> > > 3174.10888,
> > > 3062.656182,
> > > 2773.227478,
> > > 3054.042358,
> > > 3091.591142,
> > > 3053.380657,
> > > 3091.88344,
> > > 3170.547379,
> > > 3059.173669,
> > > 3172.177889,
> > > 2772.502377,
> > > 3053.160342,
> > > 3095.806169,
> > > 3174.332392,
> > > 3055.277027,
> > > 2774.841955,
> > > 3053.122873,
> > > 3089.35853,
> > > 3175.323065,
> > > 3059.886796,
> > > 2776.094817,
> > > 3062.342824,
> > > 2775.876864,
> > > 3052.111062,
> > > 3092.209838,
> > > 3171.531586,
> > > 3173.027043,
> > > 3062.680548,
> > > 3068.317677,
> > > 2772.917954,
> > > 3050.904327,
> > > 3089.180955,
> > > 2774.653639,
> > > ]
> > >
> > > DPO=[
> > > 3212.99345,
> > > 3122.570328,
> > > 3168.376866,
> > > 3050.284281,
> > > 3226.851542,
> > > 3210.994708,
> > > 3213.536831,
> > > 3113.787505,
> > > 3166.265787,
> > > 3046.372913,
> > > 3223.226982,
> > > 3216.105525,
> > > 3214.831417,
> > > 3112.141383,
> > > 3167.368775,
> > > 3047.632426,
> > > 3217.142142,
> > > 3216.914046,
> > > 3212.99345,
> > > 3122.570328,
> > > 3168.376866,
> > > 3050.284281,
> > > 3226.851542,
> > > 3210.994708,
> > > 3213.536831,
> > > 3113.787505,
> > > 3166.265787,
> > > 3046.372913,
> > > 3216.105525,
> > > 3216.105525,
> > > 3214.831417,
> > > 3112.141383,
> > > 3167.368775,
> > > 3047.632426,
> > > 3216.914046,
> > > ]
> > > ```

---

> > > > ### Comment · Reviewer_M4Kk · 2021-08-14
> > > > **Thanks for clarification on CIs**
> > > >
> > > > Thanks, I missed that these were SD not SE or CI, of course it makes sense that with n=35 the CI will be tighter than the SD. In light of this I've increased my score. I do find the $\pm$ notation a bit confusing for SD, perhaps you can emphasize this in the caption by writing e.g. $\mu \pm \sigma$? It does also seem worth reporting the actual confidence intervals somewhere, although I'm not sure if it's better to include the CIs in the main paper (instead of SDs) or in the appendix.

---

> ### Author Response · Authors · 2021-08-14
> **Thanks for your work and we are willing to address further concerns**
>
> We sincerely appreciate your responsible reviews and your constructive discussions with us, and we would like you to know that your questions provide considerably helpful guidance to improve the quality of our paper.
>
> If we may briefly summarize the discussions that have taken place so far as follows:
> - **The first is the theoretical part**. As you pointed out, some propositions or definitions are not rigorously stated. Following your comments, we carefully revised our statement for them, provided detailed proof for every proposition, and thus made our claims in the paper more sound, as you later replied.
> - **The second is for the performance in our experiment**. As mentioned by both you and other reviewers that there are concerns about the overlaps between DPO and baselines regarding the confidence interval. In light of your suggestion, we expand the number of evaluation seeds and get a more precise evaluation; we further conduct t-test to show our results are significant and explain we were actually reporting the standard deviation instead of the confidence interval itself.  This may fix your concern about the reported results. In addition, your suggestion about emphasizing the $\mu\pm\sigma$ in the caption is feasible to make the results clear.
> - For minor points, we thank you for providing several valuable questions for us to further polish the paper. In our response, we clarified these statements and will put the changes into our paper.
>
> Therefore, we hope our replies have resolved all the issues you posed and showed the improved quality of the paper. **We are always willing to answer any of your concerns** about our work and we sincerely wish you to value the technical innovation and overall contributions of the paper.

---

> ### Author Response · Authors · 2021-08-19
> **Ask for good guidance to improve our work**
>
> Dear reviewer,
>
> We first thank you again for your responsible work and valuable suggestions. And from the several rounds of discussion, we answered your questions point by point, overall we feel that we have explained in detail and addressed all of your concerns. However, we notice that you still tend to reject our paper, thus we are eager to know your remaining concerns and sincerely willing to conduct further discussions with you.
>
> Best wishes!
>
> The authors

---

### Official Review · Reviewer_wdvB · 2021-07-16

**Rating:** 7
**Confidence:** 4

**Summary:**

The paper presents a method for state-only imitation learning (SOIL) that decouples learning the expert state-transition distribution from the actual policy needed to reach desired states. In particular, given a dataset of expert state sequences, the method trains two models: one that predicts next state given the current state, and another one that learns inverse dynamics, i.e. which action is needed to reach the next state from the current state. Where the state prediction can be directly learned from the state-only expert data, environment interactions are collected using a sampling policy to learn the inverse dynamics model. Additional techniques, such as multi-step optimization and cycle training are used to improve prediction accuracy of the state-transition model. In addition, training is improved by introducing a discriminator with a GAN-like reward that can provide intermediate reward signals to the policy. The method is evaluated on several MuJoCo continuous control environments, where it is shown to outperform other SOIL baselines, and on a real-world human driving dataset where it can learn effective driving policies just from human driving trajectories.

**Limitations And Societal Impact:**

The authors discussed limitations of their work. The proposed approach does not lead to any obvious negative societal impacts.

**Main Review:**

Strengths
- State-only imitation is important for initializing robotic policies without access to ground truth actions, especially when imitating non-robotic agents like humans.
- The paper is well-written, and straightforward to understand and follow. Mathematical derivations are coherent.
- The method is shown to outperform multiple SOIL baselines and work with a real-world dataset.

Criticism
- One of the motivations for the presented work, which is discussed in Section 2, is that there is often not a one-to-one correspondence between expert state transitions and the underlying policy. However, as described later in the paper, both state-transition function and the inverse dynamics model are deterministic, which prevents them from representing a multi-modal policy. It would be interesting to see a discussion how the ambiguity of the underlying expert policy can be represented within this framework.
- To better understand scalability of the presented approach, it would be interesting to see evaluations on higher-dimensional problems, e.g. using visual inputs.

**Time Spent Reviewing:**

3

---

> ### Author Response · Authors · 2021-08-10
> **Response to Reviewer wdvB**
>
> We appreciate your constructive comments and helpful criticism.
>
> **Q1: "both state-transition function and the inverse dynamics model are deterministic, which prevents them from representing a multi-modal policy"**
>
> **A1:** We appreciate your concern, but we would like to discuss more about the "deterministic" problem. In fact, as the response A2 for reviewer GQAi, the methodology itself supports stochastic environments and stochastic policies. In addition, the word "multi-modal" is different from "stochastic" and should be treated as another specific problem. "Stochastic" means a policy can choose different actions by probabilities. However, the "multi-modal" in many cases refers to the policy distribution having more than one mode, which can be further explained as diversity/condition/task such that the actions chosen by the policy can be controlled by a preference signal (see [1] for example).
> - Below lists a detailed explanation for supporting the **"stochastic"** (almost the same as the response A2 for reviewer GQAi).
>     - Both Eq. (7) and Eq. (15) support stochastic settings where we can do reparameterization over the parameterized expectation.
>     - The specific optimization objective (along with the regularizations) accounts for different "stochastic" settings. In this paper, since we model both parts of the decoupled policy as Gaussian distributions, it can be applied to Gaussian experts and Gaussian transitions. In addition, if we want to learn in a discrete stochastic env, we can apply cross-entropy loss for the state-transition predictor. But the Gaussian choice is a common one.
>     - We have modified the conclusion part, i.e., Sec 6 to include our clarification and limitation about the above deterministic/stochastic considerations.
> - In addition, if we consider the **"multi-modal"** problem, it would be different from stochastic as we claim before. This is actually an ongoing future work where we take the "preferences" into consideration of modeling both the state transition and the inverse dynamics model.
>
> **Q2: "evaluations on higher-dimensional problems, e.g. using visual inputs."**
>
> **A2:** Thanks for your helpful advice. We think our algorithm has the potential to be applied to visual settings with more complicated and large networks like CNN to learn a well latent representation model and conduct both modules of the decoupled policy on the latent space instead. However, those modifications are non-trivial and are beyond the scope of this paper. Thus we leave this to future work and add a brief discussion in the conclusion of this paper.
>
> **More**:
>
> - We have supplement additional analysis on the toy experiment and get more interesting conclusions for our method, which can be referred to in the external link (https://anonymous.4open.science/r/dpo_rebuttal-2AF4/dpo_rebuttal_extra_fig.pdf).
>
> - We have modified our proposition with additional assumptions and corresponding proof following the other reviewers' suggestions to make the paper more rigorous and clearer, which can be referred to in the above link.
>
>
> **Reference:**
>
> [1] Li Y, Song J, Ermon S. Infogail: Interpretable imitation learning from visual demonstrations[C]//Proceedings of the 31st International Conference on Neural Information Processing Systems. 2017: 3815-3825.

---

> > ### Comment · Reviewer_wdvB · 2021-08-23
> > **Thanks for addressing my comments**
> >
> >
> > Thanks for addressing my comments and providing details on supporting the stochastic policy setting.

---

### Official Review · Reviewer_DMhm · 2021-07-16

**Rating:** 6
**Confidence:** 3

**Summary:**

This paper tackles the state-only imitation learning problem by decomposing it into two components: a state-transition predictor and an inverse dynamics model. Specifically, it argues that the problem is difficult because there are potentially multiple policies that could match the same state trajectory of the expert.

Instead, it proposes to learn what they call a hyper-policy, which is a family of policies that lead to the same distribution over state trajectories. Then, after recovering the optimal hyper-policy, it learns an inverse dynamics model to recover the actions, and this model can be learned by interacting with the environment.

However, to actually achieve good performance in practice, they propose two regularizations, i.e., a multi-step prediction loss and a cycle training loss. The parameters of the hyper-policy and the inverse dynamics model are also trained with a policy gradient objective where the reward is based on a discriminator (which is trained to predict whether the transition (s, s’) is from the expert).

**Limitations And Societal Impact:**

Yes. See the main review for other limitations.

**Main Review:**

Originality:
- This paper decomposes the state-only imitation learning problem into learning a state predictor and an inverse dynamics model. As noted by the related work, this decomposition has previously been studied in the context of imitation learning [16] and reinforcement learning [3].

- Notably, the QSS work also shows that their method can be adapted to the state-only imitation learning setting. It would be good to clarify the differences and to include a comparison in the experiments if possible.

Quality:
- The method is compared to a couple of prior methods. Comparatively, their full method demonstrates better performance. However, it seems like the reward from the discriminator is what enables good performance, more so than the decoupled structure.

- The evaluation is on Mujoco continuous control tasks and a simulated driving task. These have relatively low-dimensional state-action spaces and relatively deterministic dynamics, which makes learning a forward dynamics model, required by the cycle training, easy. I’m curious how well this would scale to higher-dimensional observations however.

Clarity:
- The paper is overall well-written and organized, and provides a good overview of the existing works in the literature in the related work section and through Table 1.

Significance:
- Imitation learning from observations is an important problem setting with real-world applications (as shown through the driving experiment).

- Since the results are quite close and the differences less discernible in many of the Mujoco tasks, evaluation on a set of harder tasks would be more convincing.


==========================

Thanks for the detailed response, and for the additional comparison to QSS.

**Time Spent Reviewing:**

3

---

> ### Author Response · Authors · 2021-08-10
> **Response to Reviewer DMhm**
>
> We thank you for your advice, and the detailed responses regarding each problem are listed below.
>
> **Q1: "Notably, the QSS work also shows that their method can be adapted to the state-only imitation learning setting. It would be good to clarify the differences and to include a comparison in the experiments if possible."**
>
> **A1:** Thanks for the valuable suggestion. The contribution of QSS is the learning of the QSS function, but we adopt the traditional Q(s, a) function. Although the QSS paper included an experiment of learning from fixed datasets pre-collected by policies with various randomness, the datasets have to contain the environmental reward so that the Q function can be directly learned. That makes the experiment more like offline RL rather than the imitation learning setting tackled by DPO.
>
> However, we value your helpful suggestion and we run QSS experiments in the rebuttal session. In our evaluation, we train QSS using the demos with the **true reward**, we run multiple hyperparameters to get the best one and train with 5 seeds, but surprisingly, we find it does not work at all. Some of the results are shown below. This makes sense after we find that the main results from the QSS paper are not such good as well. Therefore we consider not to put it in our main paper.
>
> | Hopper | HalfCheetah |
> |:----------:|:----------:|
> | 1.559 ±0.853 | -602.317±3.064  |
>
> **Q2: "the reward from the discriminator is what enables good performance, more so than the decoupled structure."**
>
> **A2:** We appreciate your concern, and we think there may still be lots of space to discuss.
> As seen in Table 2 and Figure 5, GAIfO-DP results show that only using the reward from the discriminator with decoupled policy makes similar results as normal GAIfO.
> Indeed, the reward from the discriminator plays an indispensable role in more complex experiments like Mujoco. However, thanks to the direct supervision and additional regularizations applied to the doubled structure, DPO has the potentials to enjoy higher sample efficiencies and better performances than previous methods.
> Note that the proposed regularizations can only be applied to the decoupled policy structure rather than conventional state-to-action mapping policies.
>
> **Q3: "I'm curious how well this would scale to higher-dimensional observations however."**
>
> **A3:** Thanks for your valuable suggestions! In fact, our algorithm has the potential to be applied on environments with high-dimensional, e.g. visual observations with more complicated and large networks as CNN to learn a good latent representation model and conduct both modules of the decoupled policy in the latent space instead. However, those modifications are non-trivial and are beyond the scope of this paper. Thus we leave this to future work and add a brief discussion in the conclusion of this paper.
>
> **More**:
>
> - We have supplement additional analysis on the toy experiment and get more interesting conclusions for our method, which can be referred to the external link (https://anonymous.4open.science/r/dpo_rebuttal-2AF4/dpo_rebuttal_extra_fig.pdf).
> - We have modified our proposition with additional assumptions and corresponding proof following the other reviewers' suggestions to make the paper more rigorous and clearer, which can be referred to the above link.

---

### Official Review · Reviewer_GQAi · 2021-07-17

**Rating:** 6
**Confidence:** 3

**Summary:**

This paper proposes a decoupled policy approach to state-only imitation learning, wherein the policy is decomposed into a next-state prediction network, which aims to predict the desired next state given the current state under the expert policy, and an inverse dynamics module, which given the current state and the desired next state, predicts the action. The utility of this split is that it allows for additional supervision during policy optimization, as the state transition network can be trained to match the observed expert transitions, and the inverse dynamics module can be trained on the transitions in the trajectories obtained during policy optimization. These supervised losses are placed on top of a standard policy gradient method, as gradients to both components of the decoupled policy can also be obtained through a standard likelihood-ratio policy gradient applied to a reward function corresponding to likelihood of the executed state-transitions being drawn from the expert policy. This reward function is obtained by jointly training a discriminator to classify sample transitions from the learned policy from those from the expert data.


**Limitations And Societal Impact:**

The discussion of limitations and negative societal impacts are minimal and inadequate. The limitations of an approach go beyond error bounds, but also into more practical considerations. I would encourage the authors to include a broader discussion, as I state earlier. Furthermore, while the checklist says that section 6 includes a discussion of potential negative societal impact, I do not see any discussion to this effect there.

**Main Review:**

Overall, this presents a novel approach to state-only imitation learning, which, to the best of my knowledge seems to be different from prior work primarily due to the fact that it combines state-only adversarial imitation via policy gradient (as in GAIfO) with a decoupled policy structure which allows additional supervision to each component separately. Intuitively this makes sense, and in the experiments the authors show that the approach can achieve better performance than baselines. I also appreciated the comparisons with GAIfO applied to the same decoupled network policy to show that the improved performance comes from the additional supervision, and not just the network architecture. I also appreciated the system diagrams showing both the overall framework and the regularization schemes.

The paper however also has some weaknesses in its clarity and experimental evaluation.

First, I found the organization of the paper to be confusing. The authors spend a large portion of the paper discussing the fact that the solution to the SOIL optimization problem may not be unique. While I agree with this analysis, I do not agree that this lack of uniqueness should pose problems for optimization -- gradient based optimization works well even if there might be a family of equally performing policies. The notion of "optimality" stays the same, as the authors aren't arguing that the objective of SOIL should be reconsidered. Instead, the purpose of the paper is to argue for an parameterization of the policy which takes into account the non-uniqueness of the solution.
For this reason, section 3.1 felt superfluous: The end result (equation 4) is a fairly direct conclusion from the problem statement of SOIL.

In contrast, sections 3.3 and 3.4 felt lacking in detail. In the discussion of the multi-step optimization, it was unclear why certain gradients were dropped -- the reasoning "we already have an accurate input at each time step and for each training step, we only update the first one" was not clear. Furthermore, in equation 15, it is unclear how the Q function is estimated, and also how the integral over $s'$ is computed in practice. The details in the appendix also do not provide this information. In 3.4, it is unclear what it means to "directly optimize $\mathcal{L}^\pi_{\phi,\psi}$ instead of iterative training." Also the authors state in the appendix that pre-training does not help much, and is can sometimes hurt performance, and indeed, only a few environments use a pretraining step. This should be made clear in the body of the paper, as it seems to be an important factor in performance.

The algorithm seems to be designed for deterministic environments: this is an assumption in the theorems, and the MSE-based regularization schemes do not take into account stochasticity in the dynamics or policy. While the approach seems to be effective on NGSIM environment, which is stochastic, it is evident from the appendix that neither form of regularization was used. The authors should be more transparent about this limitation of their work in the body of the paper.

Finally, it is unclear what procedure was used to tune hyperparameters for the experiments. The hyperparameters vary quite a bit from task to task, with different forms of regularization used, different forms of the reward function, different amounts of pre-training, and different amounts of gradient penalty. This suggests that the performance of the proposed method is quite sensitive to hyperparameter selection. Indeed, this may also be a flaw of prior work in this domain (as GANs are notoriously difficult to tune), but such limitations should also be more transparently discussed in the body of the paper.

Minor comments:
	- Proof of proposition 1 is not rigorous -- it should either be made rigorous, or the body of the paper be revised to say "proof sketch is in the appendix"
	- Tables should bold all entries that could statistically be the best -- for example, in Table 2, InvertedDoublePendulum, GAIfO and DPO (w PG) have overlapping confidence intervals, so bolding only DPO (w PG) is misleading.
	- In the toy experiment, it is unclear what the effect of the added action ambiguity achieves. It would be good to also include experiments comparing to k=1 to show if empirically the baseline algorithms suffer from environments with action ambiguity.


**Time Spent Reviewing:**

3

---

> ### Author Response · Authors · 2021-08-10
> **Response to Reviewer GQAi**
>
> We sincerely thank you for your comprehensive comments on our paper and we carefully answer each of your questions as below.
>
> **Q1: About the organization of the paper.**
>
> First, We apologize for the confusion of the organization in section 3. We have revised our statement in the revision to clarify most of your concerns.
>
> **Q1.1: "First, I found the organization of the paper to be confusing...For this reason, section 3.1 felt superfluous"**
>
> **A1.1:** Thanks for your valuable comments. Section 3.1 serves as a motivation to introduce our decoupled policy structure and explains why it is a better inductive bias for SOIL tasks.
>
> Furthermore, although the objective of SOIL is not changed, previous works adopt a state-to-action policy and do not analyze why and how we could resolve the SOIL task using a state-to-state mapping function, which further leads to the decoupled parameterization of the policy model.
>
> In the revision, we have renamed the section title as <Rethinking the optimality in SOIL>. Besides, according to your and other reviewers' advice, Section 3.1 is simplified and we put more space on regularizing our rigorous claim (can see in the external link (https://anonymous.4open.science/r/dpo_rebuttal-2AF4/dpo_rebuttal_extra_fig.pdf)).
>
> **Q1.2: "do not agree that this lack of uniqueness should pose problems for optimization"**
>
> **A1.2:** We appreciate your concern! We agree that previous works also work good with the uniqueness problem (this is also sensible in other research problems where under different assumptions different algorithms work), but by decoupling the policy model, we show that we find a better inductive bias for SOIL problems and thus has potential to outperform the previous methods (although with appropriate regularizations).
>
> Additionally, in our derivation and experiments, we show that the expert actions or the inverse dynamics mismatch between the agent and the expert is not the key to be modeled in SOIL tasks.
>
> **Q1.3: About Section 3.3 "unclear why certain gradients were dropped"**
>
> **A1.3:** We apologize for the unclarity. We have modified the explanation in the revision to clarify this problem:
>
> "... dropped since we already have an accurate input at each time step (the true label in the dataset) and therefore we should not require the cascade parts to be optimized using the predicted input"
>
> **Q1.4: "unclear how the Q function is estimated, and also how the integral over is computed in practice."**
>
> **A1.4:** We apologize for the confusion, and we have added additional details in our paper for clarity in our revision:
>
> "where $Q$ is the state-action value function estimated using the normal Bellman equation and proposed surrogate reward function"
>
> "In practice, Eq. (15) can be resolved via reparameterization trick. However, this can be easier in deterministic environments with deterministic expert data, where the expert state transition is a simple Dirac distribution and thus does not require the extra sampling step but can be computed directly via the output of the state transition predictor".
>
> **Q1.5: "In 3.4, it is unclear what it means to ‘directly optimize $L_{\phi,\psi}^{\pi}$ instead of iterative training’"**
>
> **A1.5:** We apologize for the unclarity. Since we have two modules, we could choose to optimize them iteratively or jointly. In our paper, we choose the latter. To ease the confusion we clarify our statement as "directly optimize $L_{\phi,\psi}^{\pi}$ instead of iterative training the two modules independently".
>
> **Q1.6: "Also the authors state in the appendix that pre-training does not help much, and is can sometimes hurt performance, and indeed, only a few environments use a pretraining step. This should be made clear in the body of the paper, as it seems to be an important factor in performance."**
>
> **A1.6:** We thank you very much for your valuable suggestion. First, in our experiment, we do not fine-tune the number of pretrain steps. In addition, as we claim, the final performances are almost the same with/without pretraining, but the training efficiency in some environments may decrease. Overall, we think it is a less important hyperparameter, and in our revision, we will take your advice to put it into the main context.
>
>
> **Q2: "...designed for deterministic environments..." "do not take into account stochasticity"**
>
> **A2:** We appreciate your concern. Actually, the methodology itself supports both stochastic environments and stochastic policies. Below lists a detailed explanation.
> - Both Eq. (7) and Eq. (15) support stochastic settings where we can do reparameterization over the parameterized expectation.
> - The specific optimization objective (along with the regularizations) accounts for different "stochastic" settings. In this paper, since we model both parts of the decoupled policy as Gaussian distributions, it can be applied to Gaussian experts or Gaussian transitions. In addition, if we want to learn in a discrete stochastic env, we can apply cross-entropy loss for the state-transition predictor. But the Gaussian choice is a common one.
> - We have modified the conclusion part, i.e., Sec 6 to include our clarification about the above deterministic/stochastic considerations.
> - Theorems are assumed to be set in a deterministic context to provide a simplified and clear conclusion about the compounding error of the decoupled structure although we claim it has many advantages.
>
> **Q3: "unclear what procedure was used to tune hyperparameters for the experiments" "hyperparameters vary quite a bit from task to task"**
>
> **A3:** We appreciate your concern, but we would like to discuss more about the "sensible" problem.
> The **only additional hyperparameters** compared with the baseline GAIfO are for the regularizations. The other ones hold the same as GAIfO. We first tune GAIfO to reach the best performance and we then evaluate different regularizations since no regularization will be hurt from the additional supervision.
> We have put the hyperparameter tuning details in Appendix D.3.
> In fact, **without fine-tuning, DPO can achieve better training efficiency than the baselines in most of the environments (see Fig. 11)**. The fine-tuning is to dig the **best potential** of our method.
>
> **Q4: "Proof of proposition 1 is not rigorous"**
>
> **A4:** Thanks very much for your detailed checking, as Reviewer M4Kk also mentions. We have clarified all the confusion of Reviewer M4Kk and we give a complete proof for proposition 1 for clarity (can be seen in the above link).
>
> **Q5: "bold all entries that could statistically be the best...so bolding only DPO (w PG) is misleading"**
>
> **A5:** Thanks very much for your careful checking. Follow your advice, we conduct evaluation experiments with more seeds (the results are updated as shown in the above link. Particularly, for InverseDoublePendulum we bold all obviously overlapped methods. For hopper, walker, and halfcheetah, we conduct a t-test to evaluate whether our method is statistically the best. The p-values against various baselines are shown below.
>
> p-value table of DPO against baselines
>
> | Env | Baseline | p-value |
> |:----------:|:----------:|:----------:|
> | Hopper | BCO  | 7.2e-09 |
> | Hopper | GAIfO  | 6.93e-10 |
> | Hopper | GAIfO-DP  | 2.29e-05 |
> | Walker | BCO  | 2.48e-24 |
> | Walker | GAIfO  | 4.01e-06 |
> | Walker | GAIfO-DP  | 0.000146 |
> | HalfCheetah | BCO  | 4.22e-30 |
> | HalfCheetah | GAIfO  | 0.000206 |
> | HalfCheetah | GAIfO-DP  | 5.35e-06 |
>
>
> Since all p-values are less than 0.05 and even are less than 0.01, we can say that our method is statistically significantly better than others.
>
> **Q6: "include experiments comparing to k=1 to show ..."**
>
> **A6:** Thanks for your valuable advice, we have now supplemented such experiments. The figures of the additional results are shown in the above link, the conclusions that will be put in our revised paper are:
> - From the experiments with k=1, BCO and DPO share similar asymptotical performance (KLD), but DPO has a significantly faster convergence rate. GAIfO also fails to find the second path.
> - In addition, we also complement two figures to visualize how well the state prediction is learned in the toy example and how the inverse dynamics of the agent is different from the expert's, which support our claim that any inverse dynamics which cover the transition support of the expert can be used to construct the expert hyper policy. Therefore, we do not have to care about the agnostic action information.
>
> **Q7: "limitations and negative societal impacts"**
>
> **A7:** We thank your suggestions and we will definitely make an additional section for discussing the societal impacts in the revision.
>
> **More**:
>
> - We have supplement additional analysis on the toy experiment and get more interesting conclusions for our method, which can be referred to the above external link.
>
> - We have modified our proposition with additional assumptions and corresponding proofs following other reviewers' suggestions to make the paper more rigorous and clearer, which can be referred to in the above link.

---

### Public Comment · Authors · 2022-03-14
**New version of paper: https://arxiv.org/abs/2203.02214**

We have reformatted the method, simplified the algorithm, and made convincing experimental results in the new version of this paper, where we illustrate how we learn a transferrable state planner and how we do transferring by pre-training. For more details, please see https://arxiv.org/abs/2203.02214.

---

### Decision · Program_Chairs · 2021-09-27

**Decision:**

Reject

**Comment:**

The paper presents a method for state-only-imitation learning. The key ideas are:
(a) Predicting future expert states from the current state.
(b) Using the inverse model to transition from current to the predicted state.
(c) Use of cycle consistency loss to improve the training of expert state prediction.
(d) Error bound on tackling the challenge of compounding error.
(e) Use of discriminator training to further improve performance.

Two reviewers score the paper as 6, one as 7, and the other with a score of 5 to reject the paper. The considerations are:

- Reviewers asked for a comparison to QSS and other baselines that the authors provided.
- Authors clarified the proof of their propositions.
- Authors clarified concerns about modeling stochastic distributions.
- Reviewers were split on the clarity of the writing, that authors promised to improve.

At the end of the rebuttal, M4Kk believes that updates both in terms of the writing and the experiments are substantial and therefore require another round of review. Considering the fact that the authors had addressed a majority of the concerns and the rebuttal incorporated all that would go in the revised version, I was initially inclined to recommend the paper for acceptance.

However, studying the paper in more detail revealed an issue with the lack of comparison with prior work that I mentioned in my comment to the authors. After considering the author's response my opinion is: (details in the discussion and the thread below):

The main premise of the paper is that state-only IL is hard because there are multiple policies consistent with a sequence of expert states. The paper proposes decoupling the learning into predicting expert states (i.e., what to do) and using the inverse model to follow the predicted expert states. However, as reviewer DMhm remarks, the most gain can be attributed to the use of discriminator rewards. The authors acknowledge this in the rebuttal, but also mention that additional supervision signals can be used due to the de-coupled policy structure which can improve performance/sample efficiency. The paper can thus be seen as proposing a set of ideas that encourage the imitator to follow the expert demonstrations better than a naive inverse model. In this view,

- Comparison with an improved version of inverse models that uses cycle consistency [1] is critical. It is because both cycle consistency in training of inverse model and the proposed method of using discriminator rewards (possible due to de-coupling) are both encouraging state-matching, albeit in different ways.

Further, if the proposed method outperforms [1], then the paper should be re-written to highlight that PG fine-tuning is critical (which experimentally leads to maximum gain, Figure 5) and decoupling allows this. The way the paper is written, it's hard to understand why de-coupling.

In light of missing comparison and that paper would have gone a substantial re-write to incorporate suggestions from all the reviewers, I am siding by reviewer M4Kk, "revisions are substantial enough that I think the paper would benefit from another review cycle, especially given the general lack of polish in the initial submission."

[1] Zero-shot visual imitation, ICLR 2018.